# Wood-Decaying Fungi: From Timber Degradation to Sustainable Insulating Biomaterials Production

**DOI:** 10.3390/ma16093547

**Published:** 2023-05-05

**Authors:** Camila Charpentier-Alfaro, Jorge Benavides-Hernández, Marco Poggerini, Alfonso Crisci, Giacomo Mele, Gianni Della Rocca, Giovanni Emiliani, Angela Frascella, Tommaso Torrigiani, Sabrina Palanti

**Affiliations:** 1Istituto per la Bioeconomia (IBE), Consiglio Nazionale delle Ricerche, Via Madonna del Piano 10, Sesto Fiorentino, 50019 Firenze, Italy; 2Centro Nacional de Innovaciones Biotecnológicas (CENIBiot), CeNAT-CONARE, San José 1174-1200, Costa Rica; 3Département Chimie, Faculté des Sciences et Technologies, Université de Lille, 59655 Villeneuve-d’Ascq, France; 4Istituto per i Sistemi Agricoli e Forestali del Mediterraneo (ISAFOM), Consiglio Nazionale delle Ricerche, P.Le Enrico Fermi, Portici, 80055 Napoli, Italy; 5Istituto per la Protezione Sostenibile delle Piante (IPSP), Consiglio Nazionale delle Ricerche, Via Madonna del Piano 10, Sesto Fiorentino, 50019 Firenze, Italy; 6Laboratorio di Meteorologia Modellistica Ambientale (LaMMA), Consiglio Nazionale delle Ricerche, Via Madonna del Piano 10, Sesto Fiorentino, 50019 Firenze, Italy

**Keywords:** mycelium-based composites, biomaterials, sustainable insulation, lignocellulosic feedstock, wood by-product valorization, fungal mycelium, material bio-fabrication, *Ganoderma carnosum*, *Trametes versicolor*, *Pleurotus ostreatus*, *Pleurotus eryngii*, *Fomitopsis pinicola*

## Abstract

Addressing the impacts of climate change and global warming has become an urgent priority for the planet’s well-being. In recent decades the great potential of fungal-based products with characteristics equal to, or even outperforming, classic petroleum-derived products has been acknowledged. These new materials present the added advantage of having a reduced carbon footprint, less environmental impact and contributing to the shift away from a fossil-based economy. This study focused on the production of insulation panels using fungal mycelium and lignocellulosic materials as substrates. The process was optimized, starting with the selection of *Trametes versicolor*, *Pleurotus ostreatus*, *P. eryngii*, *Ganoderma carnosum* and *Fomitopsis pinicola* isolates, followed by the evaluation of three grain spawn substrates (millet, wheat and a 1:1 mix of millet and wheat grains) for mycelium propagation, and finishing with the production of various mycelium-based composites using five wood by-products and waste materials (pine sawdust, oak shavings, tree of heaven wood chips, wheat straw and shredded beech wood). The obtained biomaterials were characterized for internal structure by X-ray micro-CT, thermal transmittance using a thermoflowmeter and moisture absorption. The results showed that using a wheat and millet 1:1 (*w*/*w*) mix is the best option for spawn production regardless of the fungal isolate. In addition, the performance of the final composites was influenced both by the fungal isolate and the substrate used, with the latter having a stronger effect on the measured properties. The study shows that the most promising sustainable insulating biomaterial was created using *T. versicolor* grown on wheat straw.

## 1. Introduction

The two recent UN Climate Change Conferences, COP26 and COP27, emphasized the urgent need for reducing carbon emissions, given the tight timeline necessary to achieve the aim of keeping global average temperatures within 1.5 °C above preindustrial levels [1]. Since the 2000s, the world has experienced a strong acceleration in material use related to intensive industrialization and urbanization around the globe [2,3]. Unfortunately, material efficiency, defined as all the opportunities for reducing the depletion of primary resources (e.g., extending the service life, recycling, reusing) has declined in this period [3,4], leading to important environmental impacts regarding greenhouse gas (GHG) emissions, water scarcity, resource depletion, climate change and waste accumulation [2,3].

Buildings and construction are among the sectors contributing the largest CO_2_ emissions globally [5]. Hence, in recent years there has been a rising interest in reducing the environmental footprint of buildings by studying different approaches such as thermal insulation, passive thermal storage, alternative envelopes, and energy-efficient designs [6,7,8]. However, the manufacturing of building construction materials accounted for 10% of the total global emissions in 2020 [5]. This means that, if initial embodied emissions are dismissed, as they are expelled within a very short period when compared to operational emissions, short-term CO_2_ reduction goals will be unachievable [7].

Thermal insulation plays an important role in achieving a building’s overall energy efficiency [9,10]. Moreover, even though insulation normally shows a low contribution in terms of weight of total building materials (less than 1%), it has been reported that it has large influence on environmental aspects such as global warming potential, abiotic depletion potential related to the consumption of nonrenewable sources, and eutrophication potential [11]. These negative effects can be reduced by using substitute sustainable materials with lower GHG emissions and productive processes with lower environmental impact [12,13,14].

Since 2007, mycelium-based composites have emerged as material solutions with diverse potential applications such as in artistic works [15], textile substitutes [16] and thermal and acoustic insulation and semi-structural construction materials [14,15,16,17,18,19,20]. Mycelium is the vegetative growth of filamentous fungi [20,21]. It has the capacity to break down and digest complex organic matter and polymeric substrates, and to bind and connect them into firmer materials [16,19,21,22]. This extraordinary capacity has opened the possibility of upcycling lignocellulosic waste and by-products from other industrial, forestry and agricultural processes and turning them into composite materials with the help of (mainly Basidiomycota phylum) fungi [19,22]. Further advantages of this type of material over their traditional synthetic alternatives include the low energy and water requirements of their production process, the low cost of raw materials [19,21], a reduced or even negative carbon footprint [22] and the final product’s biodegradability [23].

The aim of this work is to produce mycelium-based composites for insulating purposes. The different phases carried out to achieve fast mycelium growth and to characterize the composite material were (1) selection of several fungal strains, (2) evaluation of their growth in different spawn media, (3) production of various mycelium-based composites to understand the fungus-substrate compatibility, (4) analysis of their internal structure and (5) characterization of their thermal properties to evaluate the potential for thermal insulation.

## 2. Materials and Methods

### 2.1. Fungal Isolate Selection

Five different fungal isolates were considered in this study: *Trametes versicolor* (L.) Lloyd (1921) strain 3086 (purchased from DSMZ, Leibniz Institute DSMZ-German Collection of Microorganisms and Cell Cultures, Leibniz, Germany), *Pleurotus ostreatus* (Jacq.) P. Kumm. (1871) strain FPRL 40C (from FCBA, Institut Technologique Foret Cellulose Bois-Construction Ameublement, Bordeaux, France)*, Pleurotus eryngii* (DC.) Quél., (1872) strain M2603 (from Mycelia, Deinze, Belgium), *Ganoderma carnosum* Pat. (1889) and *Fomitopsis pinicola* (Sw.) P. Karst. (1881) both of which were collected in the silver fir forest of the Biogenetic Nature Reserve of Vallombrosa (Firenze, Italy). All isolates were grown in 9 mm diameter *Petri* plates with 2% potato dextrose agar (PDA), 2% malt extract agar (MEA) and modified Melin-Norkrans agar (MMN) at 22 °C, 25 °C and 28 °C for a week in the dark. Daily radial growth of the mycelium was measured for 7 days. For identification purposes, *Ganoderma carnosum* and *P. pinicola* isolates were grown on a sterile cellophane disk in PDA plates for 5 days at 25 °C in the dark. The mycelium was collected and lyophilized for DNA extraction by using the NucleoSpin Plant II kit (Macherey Nagel GmbH & Co., Düren, Germany, KG) according to the manufacturer’s instruction. Molecular identification was carried out by sequence analysis of Internal Transcribed Spacer (ITS) regions. Amplicons were purified with NucleoSpin Gel and a PCR Clean-up kit (Macherey Nagel GmbH & Co., KG), quantified using Qubit DNA BR (Broad Range) Assay Kits (Thermo Scientific™, Waltham, MA, USA) and an Invitrogen Qubit 4 Fluorometer (Thermo Scientific™, Waltham, MA, USA) and submitted for Sanger sequencing (Eurofins Genomics). Finally, the obtained sequences were blasted in the NCBI (National Center for Biotechnology Information) database to identify the most similar available sequences [24].

### 2.2. Grain Spawn Selection

Three different grain-based substrates were tested for mycelium propagation of the selected fungal isolates: millet and wheat grains and a mixture of 50% *w*/*w* millet and 50% *w*/*w* wheat grains were inoculated with the five fungal isolates described in Section 2.1 for mycelium growth. The grains were soaked in water for 18 h at room temperature, drained for 1 h in the air and placed in 200 mL containers filled to ¾ of their volume in weight (corresponding to 88 g of moist grains). The grain substrates in half of the containers were also supplemented with 1% *w*/*w* CaCO_3_. The containers with the grains were sterilized at 121 °C for 50 min. After cooling, 5 mycelial plugs (∼1 cm diameter) were placed on the substrates’ surfaces and the containers were incubated at 25 ± 2 °C in the dark for up to 15 days based on the growth rate of the isolate. Mycelium growth of each fungal isolate, from the top down, was monitored in quadruplicate until the whole container was colonized by the fungus. The mycelium-covered grains are referred to as ‘fungal spawn’ throughout the document.

### 2.3. Lignocellulosic Substrate Selection

Wheat straw and four different wood by-products (pine (*Pinus* spp.) sawdust, oak (*Quercus* spp.) shavings, tree of heaven (*Ailanthus altissima*) (Mill.) Swingle (1916) wood chips and shredded beech (*Fagus sylvatica* L., 1753) wood) were tested as lignocellulosic substrates for the selected fungal isolates. Sawdust is the finest wood byproduct, and is produced by sawing wood with a saw blade. Wood shavings are larger and thicker than sawdust and are typically produced by planing or shaving wood. Wood chips are larger than shavings and are typically produced by chipping wood with a chipper.

All substrates, except for pine sawdust, were supplemented with 10% *w*/*w* oat bran and were soaked in water for 18 h and finally hand-pressed for excess water removal. For pine sawdust a solid:liquid ratio of 1:2.5 (*w*/*w*) was used, adding water to the sawdust.

Approximately 500 g of moist substrate was placed in polypropylene zipper bags (SacO2, Belgium) and sterilized at 121 °C for 45 min. After cooling, 10% (*w*/*w*) of the fungal spawn was mixed with each substrate under a laminar flow hood. The bags were incubated at 25 ± 2 °C in the dark until the mycelium had colonized the material completely, then the substrate was kneaded and introduced into 9 × 12 × 4 cm molds for 4 to 5 days’ incubation in the same conditions. After this time, the substrate was removed from the mold, and oven dried at 60 ± 2 °C for 48 h to stop the mycelium growth. Bulk density was obtained for each sample using the material’s dry weight and dimensions measured using an analytical balance with three decimal places and a digital caliper, respectively. The bulk density was obtained using the ratio of the weight to the volume of the biomaterials. In addition, X-rays (50 kV, 3 mA for 8 min) of each substrate were taken to evaluate the material’s homogeneity. The X-ray analysis permitted estimation of the the difference in density by considering the different materials on gray scale, passing from white (high density material) to gray (low density material), and finally black (void space). The density can be evidenced from the X-ray image based on the difference in colour, from black to white, due to the intensity of X-rays that cross the materials depending on their different radiopacity [25].

### 2.4. X-ray Micro-CT (Computed Tomography)

A microtomograph (mod. 1272, Bruker Corporation, Billerica, MA, USA) (https://www.bruker.com/en/products-and-solutions/diffractometers-and-x-ray-microscopes/3d-x-ray-microscopes/skyscan-1272.html URL accessed on 1 April 2023) was used for internal structure imaging of the biomaterial. Nine cylinder-shaped specimens (∼2 cm diameter, ∼2 cm height) of different combinations of substrates and fungus were scanned (Table 1) using a protocol based on a low power setting (40 kV) of the cone beam X-ray microfocus source, without the use of filters, for the elimination of the low energy component of the radiation spectrum. X-ray projections (radiographs) of the sample at different rotation angles, with angular steps of 0.2°, were obtained as images at 16-bit gray levels via a CMOS sensor equipped with a fiber optic scintillator. From the radiographs, 8-bit gray-level images of the cross-sections of the specimens were then reconstructed at a resolution of 4618 PPI (image pixel size: 5.5 microns) with specific back-projection software (NRecOn v.2.0, Bruker). The image processing involved the segmentation of the solid phase via binarization carried out with a gray level threshold automatically identified with the Otsu algorithm. From the superimposition of the images of the cross-sections, the volumetric models of both the solid phase of the specimens and their porous phase were obtained. Finally, we proceeded to the 3D analysis of the images of both the solid and porous phases using specific mathematical morphology procedures [24] aimed at the detailed characterization of the structure of the specimens. The morphometric parameters listed in Table 2 were determined using CtAn v.1.19 software (Bruker); the solid and pore size distribution of the specimens was determined using a “successive opening” procedure.

### 2.5. Thermal Conductivity Assessment

Thermal transmittance (***U***) was determined by a Thermoflowmeter 635-2 with probe (Testo, Italy). ***U*** identifies how many thermal Watts (***W***) pass through 1 m^2^ of wall when there is a difference of 1 °C between the surface of the inner and outer wall (***K***), and it is defined by the following formula:$$U=\frac{W}{{Km}^{2}}$$

The formula expresses that the higher the value of ***U*** the greater the heat directional transmittance through the material. This one is specific for a dimensional structure, depending on its shape and morphology, and ***U*** essentially indicates its overall thermal insulation performance. This method of detection, normally used for windows and walls, was necessary to formulate a suitable method to be able to carry out measurements on the bio-material samples, since these had an average thickness of about 4 cm. In this work the experimental reference to evaluate full thermal resistance and transmittance of samples built by mycelium, considered as brick analogue, was the UNI 9869: 1994. The measurements were carried out in dynamic conditions. These results were not intended to be used for thermal property certifications.

The method adopted consists of heating an induction plate until to 40 °C and fixing the probe for detecting the external temperature onto the plate with paper scotch tape. Once a stable temperature of 40 °C is reached, the samples are placed on the plate, waiting at least 30 min to allow their thermal conditioning. The measured U values were recorded continuously by the instrument every 2 s for 5 min. The data average for each sample was divided by its thickness (0.04 m) in order to obtain the thermal conductivity λ (W/mK) of each biomaterial.

### 2.6. Moisture Absorption Determination

Water absorption was determined for the most promising bio-composites obtained for thermal insulation, described in Section 3.3 and Section 3.4. Samples with dimensions of approximately 50 × 25 × 15 mm^3^ were oven-dried at 103 ± 2 °C for 24 h to determine their dry weight. Then, they were placed in a conditioning chamber at a temperature of 25 ± 2 °C and 75 ± 5% RH for one month. After stabilization, each sample was weighed. The moisture content (***MC***) was calculated as a mass difference percentage using the following formula:$$MC=\frac{\left(Wf-Wd\right)}{Wd}\times 100$$
where ***Wf*** and ***Wd*** represent the final weight after conditioning in the chamber and ***Wd*** is the dry weight. Cork commercial insulating panel (Corkpanel BETONWOOD, Sesto Fiorentino, Italy) with the same dimensions as the bio-composite samples was used as reference control.

### 2.7. Statistical Analysis

The variation in mycelium growth rate attributable to substrate, temperature and fungal isolate was analyzed by using a multi-way ANOVA (Section 2.1). The influence of grain spawn media, presence of CaCO_3_ and fungal isolate on mycelium growth rate was also analyzed using the same software/approach (Section 2.2). For Section 2.4 a two-way ANOVA was performed on all the parameters listed in Table 2 using the lignocellulosic material and the fungal isolates as factors without replicates. The statistical analyses were performed by using an R stat environment [26]. The statistical significance level used in this work (*p*-value) was assumed to be equal or lower than 0.05 and graphical plots were performed by ggplot2 R packages C https://cran.r-project.org/package=ggplot2 URL accessed on 20 April 2023 Version 3.4.2 (developers: H. Wickham, W. Chang, L. Henry, T. L. Pedersen, K. Takahashi [aut],. Wilke, K. Woo, H. Yutani, D. Dunnington) [27,28].

## 3. Results

### 3.1. Fungal Strain Selection

With rising interest in sustainable biomaterials development, the need to screen existing fungal biodiversity, considering substrate and growing conditions, to find the most suitable species according to the desired material application is a priority [9]. To date, fungi, mainly from the phylum Basidiomycota, have been successfully used for biomaterials production [19,29]. Species from the genera Trametes, Ganoderma and Pleurotus are useful in the production of bio-composites and leather-like materials [29,30,31,32], whereas recent studies have demonstrated that species of the genus Fomitopsis can be used for growing mycelium skin [30].

The first stage of this study consisted of evaluating the mycelium growth rate of *G. carnosum*, *T. versicolor*, *P. ostreatus*, *F. pinicola* and *P. eryngii* using three different nutrient-rich agar media (PDA, MEA and MMN) at 22 °C, 25 °C and 28 °C. The temperatures and media tested were chosen by considering typical reported conditions for growing and maintaining the selected fungal species [21,33,34,35,36,37,38,39,40,41,42,43,44,45,46,47,48], and that the general optimum for hyphal growth of wood decay fungi is 25 °C [46]. MMN agar was included as it is often used for fungal species isolation and maintenance in the laboratory. The main objective was to select the fastest growing organisms and conditions to obtain a dense mycelium mat, as well as to establish the optimal temperature for lignocellulosic substrate selection.

As shown in Figure 1, the results obtained from the multivariate statistical analysis demonstrate that temperatures above 22 °C were preferable for growing the mycelium, but there is no significant difference between the growth rate obtained when using 25 °C or 28 °C. In addition, the mycelium growth on PDA was significantly faster than that on MEA or MMN. Based on these results, a fungal growth temperature of 25 °C and the PDA medium were chosen for the subsequent experiments.

Pictures of the mycelium of the five fungal isolates, in Petri dishes with the three different media, after 7 days of incubation at 25 °C, are shown in Figure 2. Mycelium of all the isolates was more well developed and had faster growth, in terms of ability to cover the medium, when grown on PDA.

The faster-growing fungal isolates were *G. carnosum* and *T. versicolor* 3086, compared to *P. ostreatus* FPRL 40C and *F. pinicola*, as shown in Figure 3 (*p* < 0.05). These results are comparable to those reported by [26] referring to different *Pleurotus* species, *G. lucidum* and *T. versicolor* mycelium growth on MEA measured for 7 days. The authors reported *T. versicolor* to have the highest radial growth of the strains mentioned, followed by the *Ganoderma* species and, lastly, by the *Pleurotus* ones. In the present study *P. eryngii* showed the slowest growth rate, therefore this strain was excluded from subsequent experiments. In addition, it has been reported that *F. pinicola* strains grow optimally at 30 °C [29]. Considering room temperature as the most convenient condition for a future scale up of the process, the following experiments were carried out only with *G. carnosum*, *T. versicolor* and *P. ostreatus* at 25 °C.

### 3.2. Grain Spawn Selection

Spawn is a nutritious substrate (e.g., grains) previously colonized by the mycelium that is used to inoculate the substrate for the bio-composite production [49]. This is a practice inherited from the cultivation of edible, medicinal and therapeutic mushrooms [50,51]. The grain provides many nutrient-rich inoculation sites, facilitates the homogenization of the substrate–inoculum mixture, and contributes to the rapid growth of the mycelium [48]. Common substrates used in spawn production include grains such as rye, rice, wheat, sorghum, millet, and corn, as well as cotton waste [49,50,51,52,53,54,55,56], and the selection depends mainly on the material’s availability and the fungal strain grown on it.

The second stage of this study included the evaluation of three different grain spawn media (millet, wheat and a 1:1 mix of the two) for mycelium growth of the three selected fungal strains (*G. carnosum*, *T. versicolor* and *P. ostreatus*). Wheat and millet grains have been used to provide nitrogen to accelerate fungal growth during the first stage of the biomaterial’s production [57], indicating that they are not only readily available, but also a great option regarding nutritional content for spawn media. The effect of 1% *w*/*w* CaCO_3_ supplementation was also analyzed, as it is often used as a pH regulator [52].

As shown in Figure 4, results from the statistical analysis indicate that, for all fungal isolates, growth rate is significantly faster when using the mixed grains instead of millet or wheat separately. This difference may be due to the grain’s complementing mineral composition and carbohydrate availability, as both wheat and millet have similar protein (around 10 to 15%) and starch (between 60 to 70%) contents [58,59]. The use of 1% *w*/*w* CaCO_3_ generated an average delay of 0.05 cm/day on the velocity of mycelium colonization of the grains, as shown in Table 3. As the buffering action of CaCO_3_ was proven to not have a positive impact on mycelium growth enhancement at this stage of the process, it was not added for subsequent spawn production. These results may also be linked with reduced moisture availability in the grains supplemented with CaCO_3_, as the supplement may act as a desiccant and prevent the mycelium from developing properly.

In Figure 5, the growth tendency of the three previously selected fungal isolates, using the mixed grains spawn media at 25 °C, is presented. The fastest strain was *T. versicolor*, the mycelium of which took 11 days to fully colonize the grain, followed by *G. carnosum* and *P. ostreatus*, which took 13 and 15 days, respectively. This behavior is analogous to that observed with the growth of the strains on PDA. Taking this into consideration, one can also state that, even if many parameters can influence fungal growth [58], using a standard approach with typical media in Petri dishes to evaluate the growth rate of different mycelia is a feasible and easy option for comparison between various fungal isolates.

### 3.3. Lignocellulosic Substrate Evaluation

Mycelium growth density and the mycelium–substrate degree of interfacial bonding varies significantly by species and agricultural or wood by-product employed [19]. Different samples were developed, using the previously selected fungal strains and the mixed grains as inoculum, to determine the compatibility of the fungi with the selected substrates (Figure 6).

Table 4 shows the results for the specimens obtained for each fungus–substrate combination evaluated. In general, pine sawdust performed poorly regardless of the isolate used, resulting in brittle materials that crumbled easily. This can be attributed to the physical state of this substrate, as it mainly consists of fine particles (<1 mm) that the fungal hyphae could not properly bind together. However, it was observed that this issue could be overcome by permitting the fungal skin to develop on the surface of the composite, which helps to keep the small particles together and to maintain the material’s shape. Similarly, it has been reported that, when the mycelium skin is not present, the composite tends to be less dense [55].

The tree of heaven wood chips were of particular interest for this study as it is categorized as a toxic and invasive species in Europe that is linked with negative impacts, including infrastructure damage, invasive insect proliferation and human health risks, such as allergic responses [49,50]. A similar situation to the one observed with the sawdust samples was obtained with the tree of heaven wood chips and *G. carnosum* composites. The wood chip dimensions (∼4 mm), and their lack of fibrous components, hindered the capacity of fungal hyphae to completely unify the substrate, resulting in brittle materials that also crumbled easily, as reported in Table 2. The *T. versicolor* and *P. ostreatus* isolates achieved better results in terms of the final material’s integrity, yet they took longer times to colonize the same quantity of substrate than *G. carnosum*. Fungal skin growth time differences between fungal isolates have also been reported by other authors, such as Bruscato et al. (2019) [52], who noticed that *Pleurotus albidus* requires longer growing periods to produce a mycelial ‘envelope’ when compared to *Pycnoporus sanguineus* and *Lentinus velutinus*.

Even though oak wood and its by-products have been used by several authors for growing the same fungal species tested in this study [47,48,49], the oak shavings substrate generally performed poorly, except with *P. ostreatus*, which produced a homogeneous final product with a reasonable growth time. In addition, this specific substrate was difficult to hand press to achieve the desired final form, so, overall, it was not one of the best options to work with. On the other hand, shredded beech wood composites resulted in compact and resistant composites regardless of the fungal strain. These results are consistent with the reported building components, boards and blocks produced using beech wood as a substrate and different fungal species [19,48].

Wheat straw was only evaluated using *T. versicolor* because of a lack of material availability. Results using this substrate were promising, as the final composite was light and homogeneous, and the fungal colonization of the substrate was quick. Other authors have reported mycelium composites produced using wheat straw to be good thermal insulating materials [19,45]. The properties of the composites will be discussed further in the next section.

Overall, the bulk densities obtained in this study range between the expected values for as-grown mycelium composites containing forestry by-product substrates (87–300 kg/m^3^) [19]. With the production method used in this study this characteristic depends highly on the pressure applied during the mold-filling process, which is why the obtained values varied considerably, even when using the same substrate and isolate.

The differences found between the three fungal isolates and the substrates can be attributed to the substrates’ composition, pH, nutrient availability [19,21], mycelium anatomy and structure, overall growth preferences and natural habitat of the fungal species [56].

Radiographies of the composites produced using *T. versicolor* and the five different evaluated substrates, organized from left to right according to the free space present in the final product, are shown in Figure 7. The color, from black to white, represents a value of apparent density and the distribution between particles and voids.

Shredded beech wood gave the most compact material whereas the wheat straw composite appeared as the lightest one, which is consistent with the bulk density data obtained and discussed earlier. These observations contribute to the information discussed in Section 3.4 regarding the material’s properties.

The best samples (in terms of material integrity) were selected for thermal conductivity and moisture absorption tests, as the main goal of the experiment was to find suitable fungal isolate–substrate combinations for the production of insulation panels.

### 3.4. Internal Structure Characterization by X-ray Micro-CT and 3D Image Analysis

An example of 3D images of the biomaterial specimens reconstructed by the X-ray micro-CT, showing three different substrates inoculated with *Trametes versicolor*, is reported in Figure 8.

The structure of the specimens was characterized, as a whole, by determining the overall morphometric parameters reported in Table 5 and defined in Table 2. Total porosity, mean pore size, specimen specific surface and solid-phase mean thickness are shown in the graphs in Figure 9. It can be noted that the total porosity (P) of the specimens is independent of both substrate and fungal mycelium and that it ranges between 63% and 78%, whereas the average pore diameter (P size) is significantly lower (about 120 μm on average) for specimens of pine sawdust (see two-factor ANOVA results in Appendix A) compared to tree of heaven wood chips and wheat straw (about 420 μm on average). The average thickness of solid substrate (SP thick) is not significantly different between the specimens (about 70 μm on average), whereas specimens with pine sawdust substrate showed significantly higher specific surface values (SP surf/S vol) of about 310 cm^−1^, on average, compared to those based on tree of heaven wood chips and wheat straw (about 190 cm^−1^ and 150 cm^−1^, respectively). The fractal dimension (FD) of the specimen structure was also significantly larger for specimens with pine sawdust substrate.

With regard to the structure characterization of the biomaterial, the binarized images of the solid and porous phases were processed using mathematical morphology algorithms allowing the measurement of the internal size distributions. The size classes were about 15 microns wide for both the thickness of the solid substrate and the distance between the walls within the structure of the specimens. The graphs in Figure 9 show the results expressed as both quantity distribution and cumulative curves. The marked similarity of the shape of the distributions within the same substrate clearly indicates that the internal geometry of the biomaterial specimens was basically defined by the type of lignocellulosic substrate used, and much less influenced by the fungal isolate used for the inoculum.

With regard to the study of the solid phase, the biomaterials consisting of tree of heaven wood chips showed thicknesses mainly below 300 μm with two well-expressed modal values, one around 25 μm and the other at 40 μm, along with a third nested around 75 μm. For the biomaterials made of wheat straw, the thicknesses were mainly less than 170 μm, with two well-expressed modes, one around 30 μm and the other at 70 μm, as well as one nested around 120 μm. The biomaterials consisting of pine sawdust, on the other hand, showed thicknesses mainly less than 120 μm, with a distinctly leptocurtic distribution with a modal value around 35 μm and a nested one, barely detectable around 70 μm.

The study of the pore space inside the structure of all the specimens showed, first of all, a range of dimensional variability, much greater than that of the corresponding solid-phase thicknesses, and a higher degree of multimodality, not limited only to the smaller pore sizes (Figure 9). The most frequent pore sizes (modes) found in the size range below 150 μm were 20 μm, 45 μm and 75 μm for all tree of heaven wood chips specimens, and 30 μm, 70 μm and 120 μm for both wheat straw and pine sawdust specimens, but in a much greater amount in the latter substrate. The observation of the cumulative distributions in Figure 10, moreover, allows us to highlight that the maximum size of the pores found exceeds 1.5 mm in the case of tree of heaven wood chips specimens, but is less than 1.3 mm in the case of wheat straw and less than 0.7 mm in the case of pine sawdust.

Finally, the differences between the size distributions found in the different fungal isolates show solid-phase and pore volumes of the most frequent dimensional ranges that are slightly larger for specimens inoculated with *Trametes versicolor* than for those obtained with the other two fungi.

### 3.5. Thermal Conductivity Assessment and Moisture Absorption Determination

As mentioned in the Section 3.3, some of the samples produced tended to crumble easily. For this reason, the more consistent specimens (38 in total) were selected for thermal conductivity and moisture absorption determinations. Figure 11 shows thermal conductivity in relation to bulk density of the different composites produced.

As expected, at higher densities, higher values of thermal conductivity were obtained. The association between the properties of these two materials is due to the presence of air-filled empty space in the matrix and the low thermal conductivity of air (26.2 × 10^−3^ W/mK at 0.1 MPa, 300 K) [19]. Shredded beech wood, oak shavings and pine sawdust tended to behave as a paste when moistened, leaving few voids to be filled by air. On the other hand, wheat straw and tree of heaven wood chip materials’ physical states permitted more empty spaces to remain occupied by air, attributing lower values of thermal conductivity and densities in relation to the other bio-composites.

When compared to other commonly used insulating materials, such as glass wool (57 kg/m^3^, 0.04 W/mK), sheep’s wool (18 kg/m^3^, 0.05 W/mK) and kenaf (105 kg/m^3^, 0.04 W/mK) [19,54], wheat straw, tree of heaven wood chips and pine sawdust performed well in terms of thermal conductivity. As thermal insulators are usually low-density and highly porous materials [60,61,62,63], the wheat straw composites are the most promising option from the substrates tested in this study.

When analyzing specifically the differences between the fungal isolates used for the materials’ production and focusing on shredded beechwood composites, it is evident that *G. carnosum* bonded materials tend to be denser in comparison with the *P. ostreatus* and *T. versicolor* ones. This might be explained by the differences in mycelial density due to hyphae thickness and branching. It has been reported that the hyphae of *Ganoderma* species tend to be thicker (13 µm) than those of *Trametes* (1.3 µm), for example [57]. Moreover, density differences in film applications have shown that *Ganoderma* hyphae are more likely to branch and generate a more compact structure with fewer pores than those of *Pleurotus*, whose growth tends to be more lengthwise [21].

On the other hand, moisture absorption is also inherent and of special interest for composite materials, as it can influence other characteristics, such as mechanical and thermal properties, and its application possibilities [21,46]. A material’s thermal conductivity coefficient tends to increase with increasing moisture content due to the presence of water in the previously air-occupied pores, and it can also lead to faster material decay [46]. As shown in Table 6, moisture absorption did not highlight particular differences between substrates and fungal isolates. All biomaterials reached a moisture content between 9% to 11% after one month at 75 ± 5% RH, with wheat straw and tree of heaven wood chips being at the lower end of that interval. The results obtained in the present study are in line with the results reported by Tacer-Caba et al. (2020) [21], who obtained a 10% moisture absorption in composites made from oat husk and rapeseed cake using *Trichoderma asperellum* and *Agaricus bisporus* strains when evaluating a 75% RH scenario.

Volumetric swelling (***VS***) was calculated using the following formula:$$VS=\frac{\left(Vf-Vd\right)}{Vd}\times 100$$
where ***Vf*** represents the final volume after conditioning in the chamber and ***Wd*** the dry volume.

A commercial cork panel was used as the reference for moisture absorption (6.51 ± 0.07%), the value obtained is lower than that of the biocomposites considered here. This is a known issue that must be addressed to properly introduce this type of biomaterial into existing markets, yet it was not studied further during this research. On the other hand, the fact that the samples in this study were produced by manual pressing of the colonized substrate into the desired shape must be taken in consideration, as the human factor in the production process increases variability of the specimens obtained. Nevertheless, despite all the positive results obtained regarding the use of these biocomposites, it is important to remark that people appreciate the advantages of this type of material but could be reluctant to use these materials in their own homes [15].

## 4. Conclusions

*G. carnosum*, *T. versicolor* and *P. ostreatus* grown in PDA (potato dextrose agar) at 25 °C showed a well-developed mycelium and faster growth rate in comparison to *P. eryngii* and *F. pinicola*. A 1:1 mix of wheat and millet grains was found to be the best option for fungal spawn production for these isolates, rather than using the two substrates separately. Several samples of biocomposites were produced using different wood by-product and wheat straw substrates. The composite’s substrate proved to be more influential for physical properties, such as thermal conductivity and moisture absorption, than the fungal isolate used. However, specific requirements of each fungal isolate resulted in differences in the final composite obtained. Tree of heaven wood chips and wheat straw composites were the best insulating materials, as the substrates permit the presence of more empty spaces filled with air, lowering the final material’s thermal conductivity. Using tree of heaven wood chips is a highly interesting option as it has not been reported before and it is an urban invasive species. Furthermore, the study of the biomaterial specimens based on non-destructive imaging by X-ray microtomography and the use of specific three-dimensional image analysis algorithms allowed quantification, in detail, of the internal structural differences obtained as a result of the lignocellulosic substrate and fungal mycelium used.

## Figures and Tables

**Figure 1 materials-16-03547-f001:**
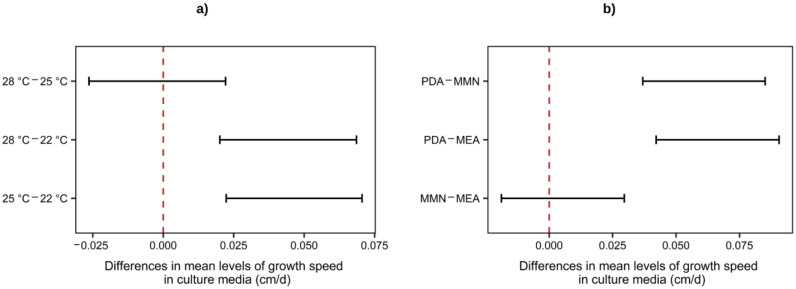
Two-way ANOVA Tukey’s range test for mycelium growth speed at different (**a**) temperatures and (**b**) culture media with a 95% confidence interval.

**Figure 2 materials-16-03547-f002:**
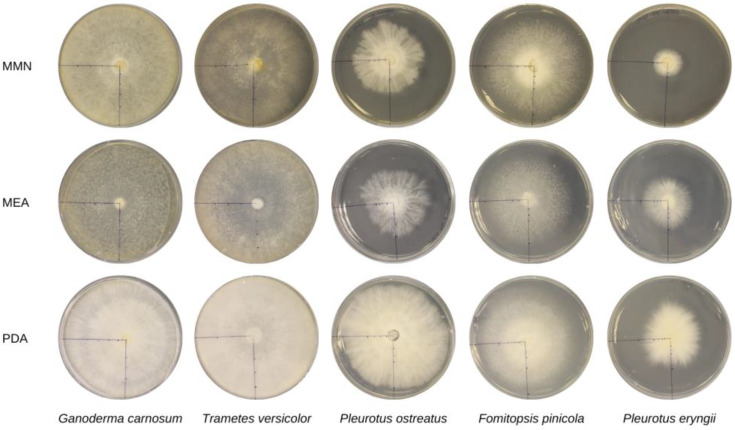
*G. carnosum*, *T. versicolor*, *P. ostreatus*, *F. pinicola* and *P. eryngii* growth on different media after 7 days of incubation at 25 °C.

**Figure 3 materials-16-03547-f003:**
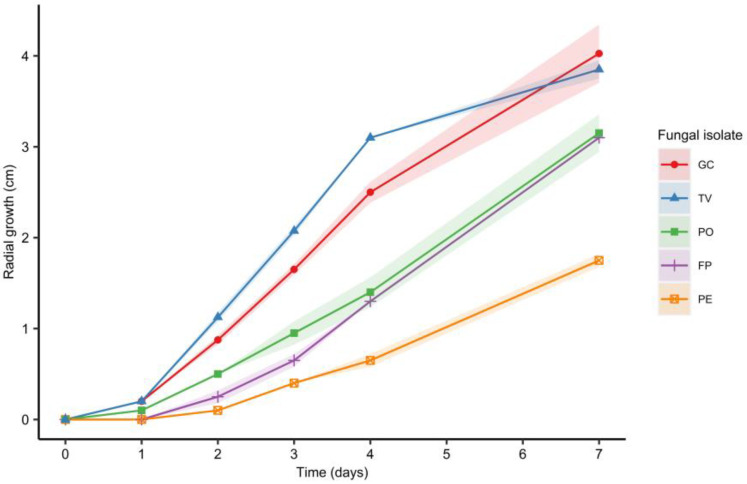
Radial mycelial growth on PDA petri dishes at 25 °C for *G. carnosum* (GC), *T. versicolor* (TV), *P. ostreatus* (PO), *F. pinicola* (FP) and *P. eryngii* (PE). Data points represent the mean and the shaded area represents two standard deviations from the mean.

**Figure 4 materials-16-03547-f004:**
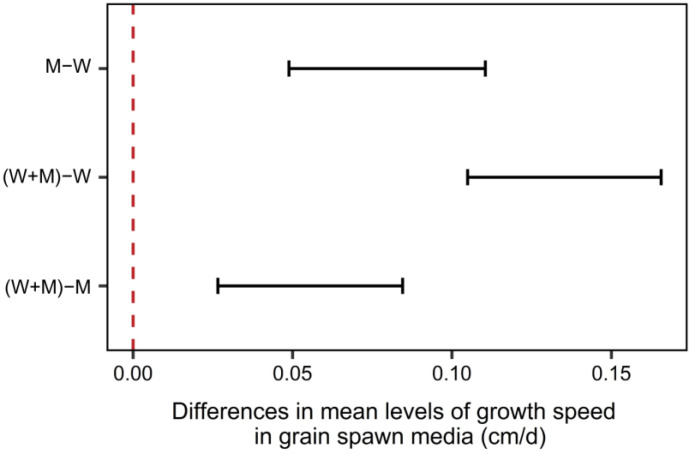
Two-way ANOVA Tukey’s range test for mycelium growth speed in different grain spawn media with a 95% confidence interval. The red dotted line means the 0.00 point.

**Figure 5 materials-16-03547-f005:**
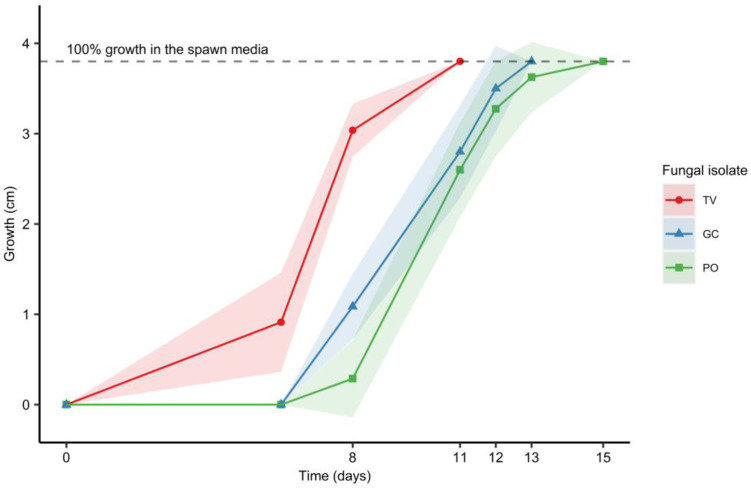
Mycelial growth trend at 25 °C on a 50% *w*/*w* millet and 50% *w*/*w* wheat grain mix for *G. carnosum* (GC), *T. versicolor* (TV) and *P. ostreatus* (PO) over time. Data points represent the mean and the shaded area represents two standard deviations from the mean.

**Figure 6 materials-16-03547-f006:**
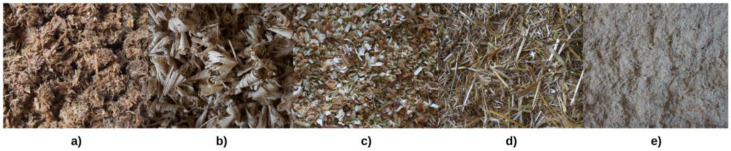
Substrates evaluated for biomaterials production: (**a**) shredded beech wood, (**b**) oak shavings, (**c**) tree of heaven chips, (**d**) wheat straw and (**e**) pine sawdust.

**Figure 7 materials-16-03547-f007:**
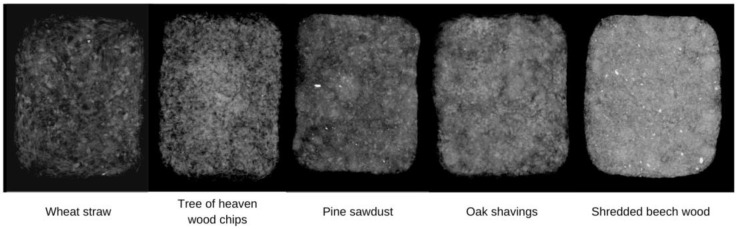
Radiographies of *T. versicolor* composites using different substrates.

**Figure 8 materials-16-03547-f008:**
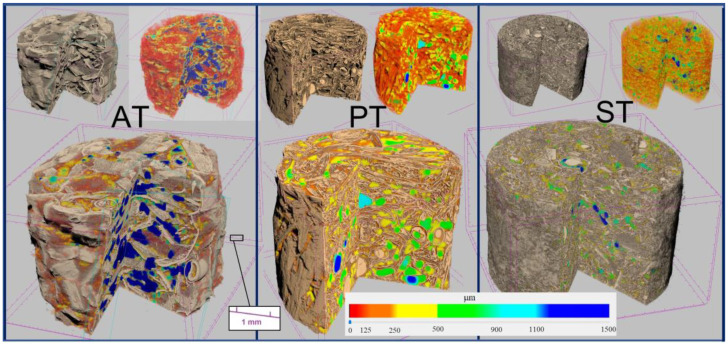
Example of 3D reconstruction of specimens imaged by X-ray micro-CT of the three different substrates inoculated with *Trametes versicolor*. AT = Tree of heaven wood chips, WT = Wheat straw, ST = Pine sawdust. For each of the three specimens, at the top left is the solid phase, at the top right is a map of the pores classified by size with a color scale and both are shown at the bottom.

**Figure 9 materials-16-03547-f009:**
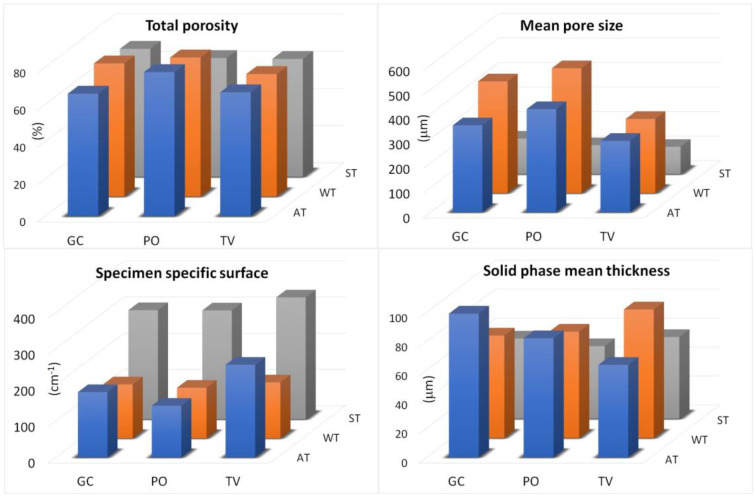
Some global structural parameters of biomaterial specimens (total porosity, mean pore size, specimen specific surface and solid-phase mean thickness). GC = *Ganoderma carnosum*, PO = *Pleurotus ostreatus*, TV = *Trametes versicolor*. AT = tree of heaven wood chips, WT = wheat straw, ST = pine sawdust.

**Figure 10 materials-16-03547-f010:**
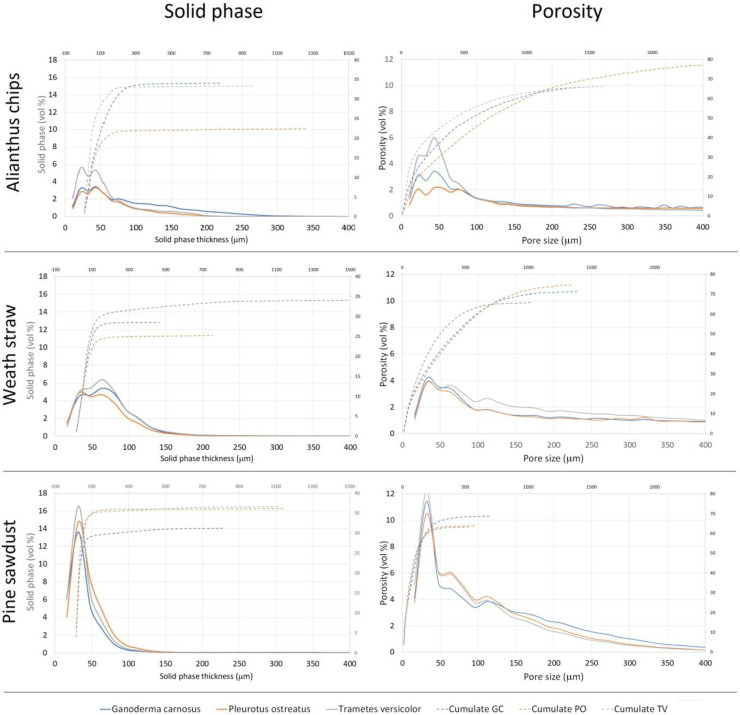
Specimen internal size distributions (**left**) of the solid phase thickness and (**right**) of the porosity of tree of heaven wood chips, wheat straw and pine sawdust substrates.

**Figure 11 materials-16-03547-f011:**
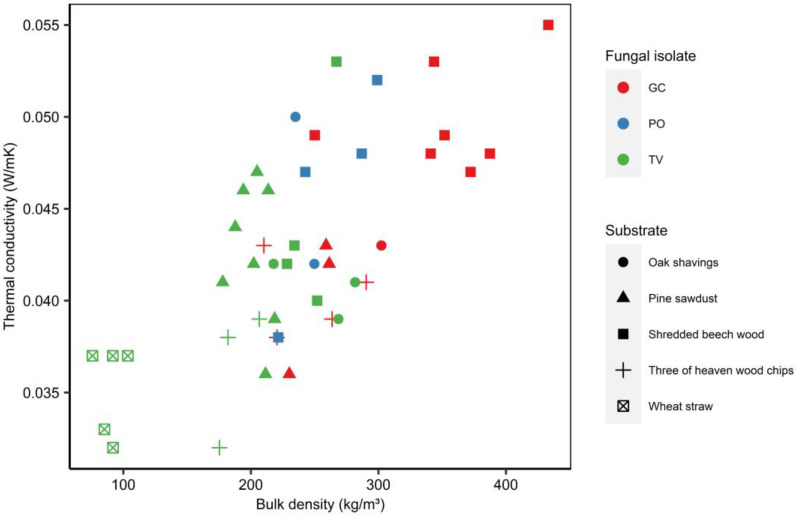
Density and thermal conductivity of mycelium composites produced using various substrates and fungal isolates. The legend regarding the used symbols is on the right of the figure.

**Table 1 materials-16-03547-t001:** Weights of the 9 specimens of different substrate, A = trees of heaven chips, W = wheat straw and S = pine sawdust, in combination with the fungal isolates, G = *G. carnosum*, T = *T. versicolor* and P = *P. osteatrus*, used for X-ray micro-CT.

Specimen ID	AG	AP	AT	WG	WP	WT	SG	SP	ST
Weight (g)	0.869	0.785	0.958	0.499	0.485	0.678	1.218	1.026	1.153

**Table 2 materials-16-03547-t002:** List of morphometric parameters determined for overall characterisation of the biomaterial structure.

Parameter	Symbol	Unit	Parameter	Symbol	Unit
Specimen volume	S vol	cm^3^	Pore volume	P vol	cm^3^
Specimen surface	S surf	cm^2^	Total porosity	P	%
Solid phase volume	SP vol	cm^3^	Closed porosity	cP	%
Solid phase surface	SP surf	cm^2^	Open porosity	oP	%
Intersection surface	SP∩S surf	cm^2^	Solid phase mean thickness	SP thick	µm
Specimen vol/S volume	SP vol/S vol	%	Mean pore size	P size	µm
Solid-phase specific surface	SP surf/SP vol	cm^−1^	Fractal dimension	FD	-
Specimen specific surface	SP surf/S vol	cm^−1^	Degree of anisotropy	DA	-

**Table 3 materials-16-03547-t003:** Mycelium growth rate on a 50% *w*/*w* millet and 50% *w*/*w* wheat grain mix for *G. carnosum* (GC), *T. versicolor* (TV) and *P. ostreatus* (PO) at 25 °C.

Fungal Isolate	1% *w*/*w* CaCO_3_ Presence	Average Mycelium Growth Rate (cm/Day)
GC	Yes	0.31 ± 0.04
	No	0.38 ± 0.04
TV	Yes	0.25 ± 0.07
	No	0.29 ± 0.03
PO	Yes	0.24 ± 0.04
	No	0.29 ± 0.04

**Table 4 materials-16-03547-t004:** Mycelium colonization times on different substrates and final composite bulk density.

Substrate	Fungal Isolate	Growth Time until Full Colonization in Bag (Days)	Bulk Density (kg/m^3^)	Observations on Homogeneity of the Final Composite
Pine sawdust	TV	25	204.6 ± 11.0	Results in brittle composites if the fungal skin is not permitted to grow on the surface
	GC	12	255.8 ± 17.1	
Wheat straw	TV	15	86.5 ± 9.6	Homogeneous composite
Oak shavings	TV	15	267.9 ± 21.4	Brittle composite
	GC	20	301.7 ± 30.3	Brittle composite
	PO	15	251.4 ± 25.0	Homogeneous composite
Tree of heaven wood chips	TV	20	186.4 ± 23.5	Homogeneous composite, better if fungal skin is grown in the surface
	GC	15	243.6 ± 27.4	Brittle composite
	PO	34	193.4 ± 18.2	Homogeneous composite
Shredded beech wood	TV	31	253.3 ± 18.0	Really compact and homogeneous composites
	GC	15	364.4 ± 21.9	
	PO	25	262.5 ± 36.5	

**Table 5 materials-16-03547-t005:** Overall structural parameters calculated for specimens of biomaterial.

Parameter	Unit	Specimen								
		AG	AP	AT	WG	WP	WT	SG	SP	ST
S vol	cm^3^	3.21	3.66	3.84	2.72	2.95	3.24	3.20	3.19	3.12
SP vol	cm^3^	1.09	0.82	1.28	0.78	0.75	1.11	1.00	1.16	1.14
SP vol/S vol	%	34.11	22.50	33.37	28.49	25.27	34.16	31.23	36.19	36.65
S surf	cm	16.45	15.01	21.62	29.31	29.47	16.97	12.80	16.91	12.45
SP surf	cm^2^	577.74	521.08	612.96	404.96	411.09	499.34	964.20	958.73	1046.76
SP∩S surf	cm^2^	4.41	1.49	3.21	3.49	2.99	4.59	2.41	2.46	3.08
SP surf/SP vol	cm^−1^	528.33	633.60	766.45	522.23	551.41	451.34	963.40	829.33	916.80
SP surf/S vol	cm^−1^	180.23	142.56	255.79	148.80	139.36	154.17	300.86	300.14	336.03
SP thick	µm	99.14	82.12	63.77	70.62	73.34	88.65	55.22	49.94	56.23
P size	µm	354.07	620.57	290.64	455.23	508.78	302.21	144.32	117.14	110.32
P vol	cm^3^	2.11	2.83	2.56	1.95	2.20	2.13	2.20	2.04	1.97
P	%	65.89	77.50	66.63	71.51	74.73	65.84	68.77	63.81	63.35
cP	%	0.38	0.56	0.61	0.45	0.41	0.29	0.10	0.08	0.09
oP	%	65.76	77.37	66.42	71.38	74.62	65.74	68.74	63.78	63.31
FD	-	2.68	2.68	2.81	2.71	2.70	2.80	2.88	2.90	2.91
DA	-	1.29	1.30	1.41	1.58	1.16	1.31	1.12	1.09	1.08

**Table 6 materials-16-03547-t006:** Moisture absorption and volumetric swelling coefficient of mycelium composites produced using various substrates and different fungal strains after one month at 25 ± 2 °C and 75 ± 5% RH.

Substrate	Fungal Strain	Moisture Absorption (%)	Volumetric Swelling (%)
Pine sawdust	TV	10.79 ± 0.15	5.43 ± 0.49
	GC	10.96 ± 0.11	2.20 ± 0.73
Wheat straw	TV	9.67 ± 0.14	1.78 ± 1.64
Oak shavings	TV	10.56 ± 0.18	0.77 ± 0.30
	GC	11.08 ± 0.11	0.28 ± 0.31
	PO	10.63 ± 0.68	1.41 ± 0.41
Ailanthus chips	TV	9.06 ± 0.03	6.57 ± 0.93
	GC	9.55 ± 0.23	3.30 ± 0.78
Shredded beech wood	TV	9.85 ± 0.28	3.88 ± 0.82
	GC	10.56 ±0.13	2.20 ± 1.89
	PO	10.72 ± 0.19	1.20 ± 0.52

## Data Availability

All data supporting this investigation’s conclusions are available from the corresponding author.

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
