# Peer review of "Wood-Decaying Fungi: From Timber Degradation to Sustainable Insulating Biomaterials Production"

_materials, 2023, doi:10.3390/ma16093547_

Round 1

Reviewer 1 Report

Dear Authors

The reviewed study focuses on the production of insulation panels using fungal mycelium and lignocellulosic materials as a substrate. As it is well-known in recent decades, there has been an acknowledgment of the great potential of fungal-based products, which possess similar or superior characteristics to classic petroleum-derived products. The advantage of these new materials is their reduced carbon footprint, less environmental impact, and potential to shift away from a fossil-based economy, making it an essential priority at the planetary level to contrast climate changes and global warming. The study optimized the process by selecting isolates from Trametes versicolor, Pleurotus ostreatus, Pleurotus eryngii, Ganoderma carnosum, and Fomitopsis pinicola, evaluating three-grain spawn substrates (millet, wheat, and a 1:1 mixture of millet and wheat grains) for mycelium propagation, and producing variants of mycelium-based composites using five wood byproducts and waste materials (pine sawdust, oak shavings, tree of heaven wood chips, wheat straw, and shredded beech wood). The biomaterials' internal structure was characterized by X-ray micro-CT, thermal transmittance using a thermoflowmeter, and moisture absorption. The results showed that using a 1:1 mixture of wheat and millet is the best option for spawning fungi, regardless of the fungal isolate. The final composites' performance was influenced by the fungal isolate and the substrate used, with the substrate having a more substantial effect on the measured properties. The study demonstrates that the most promising sustainable insulating biomaterial creates Trametes versicolor grown in wheat straw.

I find the work interesting and worth publishing, but I have some comments to consider by the authors:

1.       Line 14. "The necessity of contrasting climate change and consequent global warming is a priority at planetary level"? Maybe "Addressing the impacts of climate change and global warming has become an urgent priority for the planet's well-being" would be more concise.

2.       Lines 30-31 Instead "From the mycelium-substrate couples evaluated, the most promising insulating material was obtained when growing T. versicolor in wheat straw" better "The study shows that the most promising sustainable insulating biomaterial creates T. versicolor that grows in wheat straw".

3.       Line 57. Please remove the space before the "%" sign.

4.       Line 63. I suggest expanding the statement: "In recent years, mycelium-based composites have emerged as interesting material solutions with diverse and disruptive potential applications, including thermal and acoustic insulation and semi-structural construction materials [14–18]". Maybe the expanded statement "Since 2007, mycelium-based composites have emerged as material solutions with diverse and disruptive potential applications such as artistic works [10.3390/ma16062164], textile substitutes [Gandia, A.; van den Brandhof, J.G.; Appels, F.V.W.; Jones, M.P. Flexible Fungal Materials: Shaping the Future. Trends Biotechnol. 2021, 39, 1321–1331] and thermal and acoustic insulation and semi-structural construction materials [14–18]” the authors consider more complete.

5.       Line 102. Why "NCBI database"? (I assume that NCBI = National Center for Biotechnology Information, because the abbreviation is not explained, and the reference to the cited database is missing). I suggest referring to commonly used fungi databases containing updated information on fungi, such as Species Fungorum (www.speciesfungorum.org, Centre for Agriculture and Bioscience International (CABI), Wallingford, Oxfordshire, UK), and/or MycoBank (www.mycobank.org, Westerdijk Fungal Biodiversity Institute, Utrecht, Belgium). Otherwise, please briefly justify the choice of the NCBI database and expand the acronym and add references.

6.       Line 106. Please remove the space before the "%" sign (two times). Please revise the whole manustript (lines 127, 200, 282, 292 and more).

7.       Line 120. Please explain the difference between "shavings", "chips" and "shredded" [beechwood] ("tree of heaven […] wood chips and shredded beech […]"). Maybe this is the same form of wood material of different wood species? Please consider standardizing this in the whole manuscript to one word (or phrases with one word), for example, "oakwood chips", "ailanthus-wood chips" (as on fig. 10) and "beechwood chips" (lines 24, 320, 346, 359 – table 4 – and more).

8.       Line 130. Please change "9x12x4 cm" to "9 × 12 × 4 cm".

9.       Line 144. Please change "The Bruker microtomograph mod. 1272 (http://bruker-microct.com/products/1272.htm)" to "The microtomograph (mod. 1272, Bruker Corporation, Billerica, MA, USA)".

10.   Line 153. "resolution of 5.5 m pixel"? Please provide in PPI (px per inch).

11.   Line 168. Table 2. Please remove spaces after the brackets in column "Units" (Do we need parentheses?).

12.   Line 178. Italicize "U".

13.   Line 198. Please change “50 x 25 x 15 mm3” to “50 × 25 × 15 mm3”.

14.   Line 205. To "Corkpanel BETONWOOD" add the city and country.

15.   Line 212. "Two-way ANOVA" or "two-way ANOVA"?

16.   Line 265. Please provide Figure 3 in higher resolution; change the "Radial Growth" into "Radial growth".

17.   Line 279. Italicize the fungi species' names.

18.   Line 343. Italicize the fungi species' names.

19.   Line 352. "beech substrates"? maybe "beechwood as a substrate" would be more precise?

20.   Line 362. "3" should be superscript.

21.   Line 414. Please provide Figure 9 in higher resolution, with black font (not gray). Please compare the caption with the graph; there is no "AT" in the graph. Please change "=" to "–" and add spaces.

22.   Line 430. Please provide Figure 10 in higher resolution. "Pine Sawdust" or "Pine sawdust"? The second is better.

23.   Ine 463. Please provide Figure 11 in higher resolution.

24.   Line 468. Change "x" to "×".

25.   Lines 473-474. "3" should be superscript.

26.   Lines 479-480. Italicize the fungi species' names.

27.   Line 509. Expand "PDA" into "potato dextrose agar". I think unusual, rarely used abbreviations should be consistently expanded at the beginning of each article section.

28. In my opinion, it would be worth referring in the Discussion section to a crucial aspect related to the use of engineering materials using mushrooms. I mean their level of acceptance by people. The research article [10.3390/ma16062164] stated that people appreciate the advantages of this type of material but would be reluctant to use these materials in their own homes.

 To sum up. Everything I've read is very interesting.

The authors provided a study investigating the growth and properties of several fungal isolates, including Ganoderma carnosum, Trametes versicolor, Pleurotus ostreatus, Pleurotus eryngii, and Fomitopsis pinicola. The researchers found that the first three isolates showed a well-developed mycelium and faster growth rate compared to the latter two isolates.

Additionally, the study found that a 1:1 mix of wheat and millet grains was the best option for fungal spawn production for these isolates instead of using the two substrates separately. The researchers also produced several biocomposite samples using wood byproducts and wheat straw substrates, finding that the composite's substrate had a more significant effect on physical properties like thermal conductivity and moisture absorption than the fungal isolate used.

However, the specific requirements of each fungal isolate resulted in differences in the final composite obtained. For example, according to the authors, Ailanthus (tree of heaven) wood chips and wheat straw composites were found to be the best insulating materials because the substrates provide more empty spaces filled with air, lowering the final material's thermal conductivity. Interestingly, using Ailanthus wood chips was the most profitable variant as it has not been reported before and is an urban invasive species.

Finally, the authors used non-destructive imaging by X-ray microtomography and specific three-dimensional image analysis algorithms to quantify the internal structural differences obtained depending on the lignocellulosic substrate and fungal mycelium used. The method used "provided" a very interesting Figure 8.

Unfortunately, reading the article was hindered by numerous editorial small errors, which I pointed out in the comments above. I kindly ask you to correct them because the article is worth publishing.

 Sincerely

Author Response

In the attached file the replies to revisor 1

Reviewer 2 Report

This manuscript focuses on the production of insulation panels using fungal mycelium and lignocellulosic materials as sub-19 strate. When the mycelium-substrate couples evaluated, the most promising insulat- 30 ing material was obtained when growing T. versicolor in wheat straw. This study is quite novel. However, there are still some questions in the manuscript that need to be answered:

1. In Introduction, the relationship between mycelium-substrate and thermal insulation properties of wood materials is not described, so it is suggested to add.

2. It is suggested to supplement the micro-morphology of materials.

3. There is no content related to mechanical properties. It is suggested to add.

4. It is suggested to further check the full text. There are some mistakes in details.

5. It is suggested to add the infrared thermal imaging figures during heating to show the heat insulation effect more directly.

6. It is suggested to supplement the size change of the plate during the hygroscopic test.

Author Response

IN the attacched the replies to Revisor 2

Round 2

Reviewer 1 Report

Dear dr. Sabrina Palanti,

Thank you for addressing all my comments. I have no more substantive comments, but:

1.       Lines 40-41. Please consider the use of "recent" instead of "last" and the use of "urgent need" instead of "prompt action" to convey a stronger sense of the importance of addressing carbon emissions.

2.       Lines 123-126 Please consider adding a piece of information about the morphology of wood particles, I mean the difference between "sawdust", "shavings" and "chips". It is important in the context of the topic. As you wrote in response to my comment: sawdust is the finest wood byproduct, and it is produced by sawing wood with a saw blade. Wood shavings are larger and thicker than sawdust and are typically produced by planing or shaving wood. Wood chips are larger than shavings and are typically produced by chipping wood with a chipper.

3.       Line 223. ("ggplot2 R packages") Please provide a software version, the developer name, city, and country.

4.       Line 325. Remove ":".

5.       Line 416. Remove “(A. altissima)”

6.       Line 423. Please change commas for dots and remove the brackets in the "units" column.

7.       Line 428. Remove “(Ailanthus altissima)”

8.       Line 441. Change "Ailanthus altissima (tree of heaven)" to "tree of heaven".

9.       Lines 528-529. Italicise fungi species names.

10.   Line 535. Change "Ailanthus altissima (tree of heaven)" to "tree of heaven".

Overall, the revised manuscript is a well-supported evaluation of the importance of finding sustainable alternatives to fossil fuels and the potential for fungal-based products to play a role in reducing industrial processes' carbon footprint and environmental impact. The study provides valuable insights into producing sustainable insulation panels and contributes to the growing research on using fungi in sustainable materials development.

Sincerely,

Author Response

Dear Revisor, In the attached file my replies.

Reviewer 2 Report

I have no further suggestions for modification.

Author Response

Dear Revisor thank you for your help and for accepting my paper in the last version.
